# Fibroblast Growth Factor 20 Gene Polymorphism in Parkinson’s Disease in Asian Population: A Meta-Analysis

**DOI:** 10.3390/genes12050674

**Published:** 2021-04-29

**Authors:** Han-Lin Chiang, Yih-Ru Wu, Yi-Chun Chen, Hon-Chung Fung, Chiung-Mei Chen

**Affiliations:** 1Department of Neurology, Neurological Institute, Taipei Veterans General Hospital, Taipei 112, Taiwan; yorkiego@gmail.com; 2Chang Gung Memorial Hospital Linkou Medical Center and College of Medicine, Department of Neurology, Chang Gung University, Taoyuan 333, Taiwan; yihruwu@cgmh.org.tw (Y.-R.W.); asd108@adm.cgmh.org.tw (Y.-C.C.); 3Fu Jen Faculty of Theology of St. Robert Bellarmine, Fu Jen University Clinic, Taipei 242, Taiwan; pf200272@hotmail.com

**Keywords:** Parkinson’s disease, fibroblast growth factor 20, FGF20, rs591323

## Abstract

Parkinson’s disease (PD) is a neurodegenerative disease with the pathological hallmark of Lewy bodies and Lewy neurites composed of α-synuclein. The SNP rs591323 is one of the risk loci located near the *FGF20* gene that has been implicated in PD. The variation of *FGF20* in the 3′ untranslated region was shown to increase α-synuclein expression. We examined the association of rs591323 with the risk of PD in a Taiwanese population and conducted a meta-analysis, including our study and two other studies from China, to further confirm the role of this SNP in Taiwanese/Chinese populations. A total of 586 patients with PD and 586 health controls (HCs) were included in our study. We found that the minor allele (A) and the AA + GA genotype under the dominant model are significantly less frequent in PD than in controls. The meta-analysis consisted of 1950 patients with PD and 2073 healthy controls from three studies. There was significant association between rs591323 and the risk of PD in the additive (Z = −3.96; *p* < 0.0001) and the dominant models (Z = −4.01; *p* < 0.0001). Our study results and the meta-analysis support the possible protective role of the rs591323 A allele in PD in Taiwanese/Chinese populations.

## 1. Introduction

Parkinson’s disease (PD) is a neurodegenerative disease with the pathologic hallmark of dopamine neuron loss in the substantia nigra pars compacta (SNpc) [1,2]. In addition, α-synuclein pathology, including Lewy bodies and Lewy neurites, is commonly found in parkinsonian brains; however, the exact role of these abnormal protein aggregations is still elusive and does not correlate with neuronal cell loss [3,4]. The interplay between genetic and environmental risk factors is known to contribute to the development of PD [1,2]. With the help of advanced technology, genome-wide association studies (GWASs) and meta-analyses have identified at least 43 genetic PD risk loci for European ancestry [5]. The single nucleotide polymorphism (SNP) rs591323 is one of the risk loci that have been implicated in PD in meta-analyses of GWAS studies and case-control studies [5,6,7,8,9] and is among the top 20 PD risk foci identified by GWAS meta-analysis in the PDGene database (http://www.pdgene.org/gwas (accessed on 4 April 2021).

The SNP rs591323 is located in an intergenic region in 8p22 near the *FGF20* gene [6], which is also known as a risk gene for PD [10]. Interestingly, the variation rs12720208 of *FGF20* in the 3′untranslated region was shown to increase the expression of α-synuclein [11], whereas if SNP rs591323 influences FGF20 or α-synuclein expression is not known.

In this study, we aimed to examine the association of rs591323 with the risk for PD in a Taiwanese population and conduct a meta-analysis to further confirm the role of this particular SNP in Taiwanese/Chinese population.

## 2. Materials and Methods

### 2.1. Study Subjects

Patients who fulfilled the UK PD Society Brain Bank Clinical diagnostic criteria [12] were recruited from the neurology clinics of Chang Gung Memorial Hospital by two movement disorder specialists, YR Wu and CM Chen. Controls were healthy volunteers matched for gender and ethnicity. All subjects gave informed consent. This study is approved by the institutional review board of Chang Gung Memorial Hospital (ethical license number: 102-5614A3).

### 2.2. Genetic Analysis

The FGF20 SNP rs591323 is among the top 20 PD risk foci identified by GWAS meta-analysis in the PDGene database (http://www.pdgene.org/gwas (accessed on 4 April 2021)). We used the Agena MassARRAY platform with iPLEX gold chemistry (Agena, San Diego, CA, USA) for genotyping of the 20 SNPs following the manufacturer’s protocol. We designed the sequence of the polymerase chain reaction (PCR) primers and the extension primers with the Assay Designer software package (v.4.0). In brief, a multiplex PCR reaction was performed in a total volume of 5μL that consisted of 1μL DNA sample (10 ng/μL), 1 unit of Taq polymerase, 500 nmol of each PCR primer mix and 2.5 mM of each deoxynucleotide (Agena, PCA accessory and Enzyme kit). Thermocycling was performed in the following sequence: 94 °C for 4 min, 45 cycles of 94 °C for 20 s, 56 °C for 30 s, 72 °C for 1 min, and finally 72 °C for 3 min. In total, 0.3 units of shrimp alkaline phosphatase was used to deactivate unincorporated deoxynucleotides. iPLEX enzyme, terminator mix, and extension primer mix were used for the single base extension reaction with thermocycling (94 °C for 30 s, 40 cycles of 94 °C for 5 s, 5 inter cycles of 56 °C for 5 s, 80 °C for 5 s and finally 72 °C for 3 min (Agena, iPLEX gold kit)). A cation exchange resin was used to remove residual salt from the reactions. In brief, the clean resin was spread out on the dimple plate using the scraper and the resin was left to dry for 10 min at room temperature. Each iPLEX Gold extension product was added with 16 µL high performance liquid chromatography-grade water. The sample plate was sealed and centrifuged at 3200× *g* for 1 min. The sample plate was gently inverted on the top of the dimple plate. The sample plate wells were aligned to the well of the resin, and then the both plates were inverted again so that the dimple plate was on the top of the sample plate. The dimple plate was gently tapped to let the resin fall into the sample wells and then the dimple plate was removed. The sample plate was sealed and rotated for at least 15 min at room temperature to remove the salt. The purified primer extension reaction was analyzed on a matrix pad of a SpectroCHIP using a MassARRAY Analyzer 4. The TYPER 4.0 software was used for clustering analysis.

### 2.3. Statistical Analysis

All statistical analysis was performed using IBM SPSS statistics 23. Genotypes were examined for Hardy–Weinberg equilibrium. A chi square test was used to compare allele frequencies, genotypes, dominant and recessive models between patients with PD and controls. All *p* values were two-tailed and were considered statistically significant if *p* < 0.05.

### 2.4. Meta-Analysis

#### 2.4.1. Literature Search and Inclusion

Keywords including Parkinson’s disease, FGF20, Fibroblast growth factor 20, and rs591323 were used to search in multiple databases including Pubmed, EMBASE, and Web of Science (Figure 1). Reference lists were also screened for possible inclusion. The publication was included in the meta-analysis only if it was a case-control study and if there were sufficient data including genotype frequency for data analysis.

#### 2.4.2. Statistical Analysis

We examined the effect of the SNP to the risk of PD in the additive model (G versus A alleles), dominant model (AA + AG versus GG), recessive model (AA versus AG + GG) using the odds ratio (OR) with 95% confidence intervals (CIs). Heterogeneity between the studies was determined by a Q test and I^2^. The studies were considered to lack heterogeneity if *p* value for the Q test was >0.10 and I^2^ was <50%. As there was lack of heterogeneity between the included three studies in all three models (additive, recessive and dominant), the pooled OR was calculated using the fixed effect model (the Mantel–Haenszel method). Z test was used to determine the significance of the pooled OR.

## 3. Results

Parts of the results of the top 20 risk SNPs for PD analyzed by using iPLEX methodology as described above have been published previously [13,14,15,16]. A total of 586 patients with PD and 586 healthy controls (HCs) were included for analysis of *FGF20* and rs591323 in this study. The mean age of the patients with PD was older than the HCs, but not significantly (PD: 68.58 ± 10.89, HC: 59.53 ± 12.78; *p* > 0.05). Gender distribution is similar between groups (*p* = 0.06). The distribution of all the genotypes followed the Hardy–Weinberg equilibrium (Table 1). The distribution of the rs591323 genotype frequency is significantly different between PD and HC (*p* = 0.011) and the A allele is less frequent in PD than in control (*p* = 0.029). In the dominant model, the AA + GA genotype frequency is significantly higher in HCs than in PD (69% versus 61%; *p* = 0.003). When calculated according to the recessive model, there is no difference between the two groups (*p* = 0.75) (Table 2).

### Meta-Analysis

A total of 62 publications were included in the initial search. We further reviewed the titles and the abstracts of the remaining publications, and studies were excluded if they were nonhuman, non-PD, did not include the locus rs591323 or if they were not case-control studies (*n* = 55). A further 29 studies were excluded through full-text reading to include only the SNP rs591323 and the Taiwanese/Chinese population. Finally, two studies were included for meta-analysis (Figure 1).

The heterogeneity among the three studies, our study and the two studies from China, was not significant under all tested models (additive model: *p* = 0.68, I^2^ = 0.00%; dominant model: *p* = 0.56, I^2^ = 0.00%; recessive model: *p* = 0.43, I^2^ = 0.01%). A total of 1950 PD patients and 2073 HCs were included in the meta-analysis. There was significant association between rs591323 and the risk of PD in both the additive model (Z = −3.96; *p* < 0.0001) and dominant model (Z = −4.01; *p* < *0*.0001). The results suggest a possible protective role of the A allele and GA + AA genotype in the dominant model in PD (Figure 2).

## 4. Discussion

The SNP rs591323 was first identified as one of risk loci for PD by the International Parkinson’s Disease Genomics Consortium (IPDGC) in 2011 [6]. The result was later replicated by a case-control study in a Scandinavian population [17], as well as a large-scale meta-analysis comprising populations from European countries and the USA [7]. As for the Asian population, there were three case-control studies from the Han Chinese population [8,9,18]. Jing et al. showed a significant genotype difference in rs591323 between PD patients and controls, with increased frequency of GA and AA and decreased frequency of GG in the control group [8]. The study conducted by Sun and colleagues also showed that both the A allele and the dominant model (AA + AG versus GG) of rs591323 are protective against PD [9], whilst in the study performed by Liu et al., the result could not be interpreted due to rs591323 significantly deviating from the Hardy–Weinberg equilibrium in both study groups [18]. In support of the previous studies [6,7,8,9,17,18], our study also showed a significant genotype differences between the PD and control groups (*p* = 0.011) and a reduction in PD risk for the A allele (*p* = 0.029) and dominant model (AA + AG versus GG) (*p* = 0.003). The results of a further meta-analysis support the protective role of the A allele of rs591323 as the results were significant in the additive and dominant models.

rs591323 is located in the intergenic region in chromosome 8 near the *FGF20* gene. Fibroblast growth factor-20 (FGF20) is one of the neurotrophic factors that was found to be selectively expressed in the SNpc [19], promoting dopamine neuron differentiation and survival [10]. It has been shown that *FGF20* genetic variations including one SNP located in the intron (rs1989754) and two SNPs (rs1721100 and ss20399075) located in the 3′ regulatory region are highly associated with the risk of PD [20]. Moreover, rs1721100 (C/G) polymorphism was shown to be a risk factor for PD in Japanese and Chinese populations [21,22]. However, whether or not the variation of rs591323 affects the activity of *FGF20* remains to be determined. Pak et al. investigated the associations between 19 PD risk loci and dopamine transporter (DAT) and serotonin transporter (SERT) availability in healthy controls. While none of the SNPs had an effect on DAT availability, the AA genotype of rs591323 was associated with lower SERT availability in both pons and midbrain as compared to the AG and GG genotypes [23]. Although it was shown that lower availability of SERT in the thalamus is associated with anxiety in PD [24], the clinical implication of lower SERT availability in pons and midbrain is largely unknown.

## 5. Conclusions

In conclusion, our study is the first meta-analysis to show that the A allele of rs591323 is associated with reduced risk in the Asian PD population. However, the study results should be interpreted carefully with the following limitations of the study in mind. First of all, the sample size was small and only three case-control studies were included for meta-analysis. Second, the average age of the controls was numerically younger than PDs, and conversion of controls to PD at an older age may be possible. Third, the results were not adjusted according to other factors such as patient’s age, sex, and other environmental factors due to lack of information. Lastly, the study result does not directly prove the causal relationship between this particular SNP variation and PD; other possible mechanisms, including gene–gene interactions or linkage disequilibrium with other risk factor genes, may also be involved.

## Figures and Tables

**Figure 1 genes-12-00674-f001:**
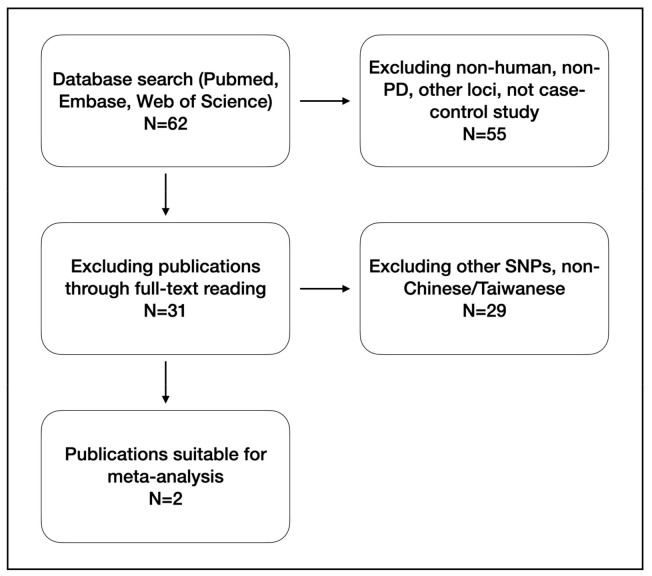
Literature search and inclusion.

**Figure 2 genes-12-00674-f002:**
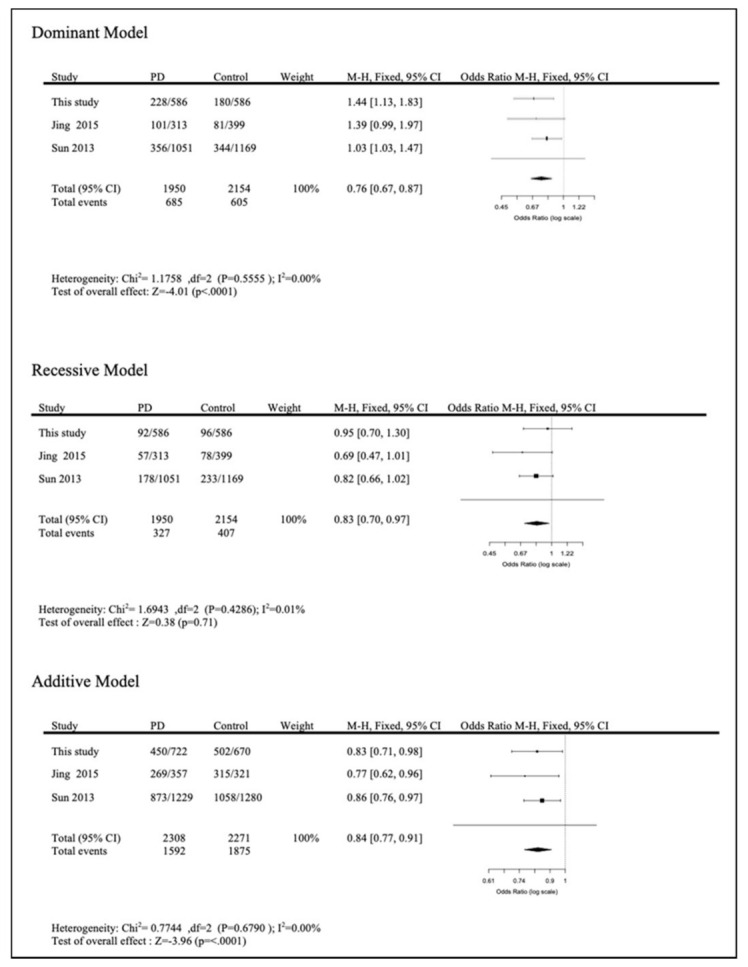
Meta-analysis of different models.

**Table 1 genes-12-00674-t001:** Genotype distribution of the three studies included in the meta-analysis.

First Author	Country	PD					Controls					HWE for PD		HWE for Controls	
		Genotype			Allele		Genotype			Allele					
		GG	GA	AA	G	A	GG	GA	AA	G	A	X^2^	P	X^2^	P
Chiang	Taiwan	228	266	92	722	450	180	310	96	670	502	0.96	0.33	3.77	0.052
Jing	China	101	155	57	357	269	81	159	78	321	315	0.035	0.85	2.52 × 10^−6^	0.99
Sun	China	356	517	178	1229	873	344	592	233	1280	1058	0.174	0.67	0.568	0.45

HWE, Hardy–Weinberg equilibrium.

**Table 2 genes-12-00674-t002:** Summary of the three studies included in the meta-analysis.

rs591323 Genotype/Allele	PD (%)	Control (%)	OR	*p* Value
This study (Taiwan) (*n* = 586 + 586 = 1172)				
GG	228 (39%)	180 (31%)		
GA	266 (45%)	310 (53%)		
AA	92 (16%)	96 (16%)		**0.011**
G	722 (62%)	670 (57%)	1	
A	450 (38%)	502 (43%)	0.832 (0.705, 0.981)	**0.0287**
Dominant model				
GA + AA	358 (61%)	406 (69%)	0.696 (0.547, 0.886)	**0.0032**
GG	228 (39%)	180 (31%)	1	
Recessive model				
AA	92 (16%)	96 (16%)	0.956 (0.696, 1.298)	0.7502
GA + GG	494 (84%)	490 (84%)	1	
Jing et al. (China) (*n* = 313 + 318 = 631)				
GG	101 (32%)	81(25%)		
GA	155 (50%)	159 (50%)		
AA	57 (18%)	78 (25%)		
G	357 (57%)	321 (50%)	1	
A	269 (43%)	315 (50%)	0.768 (0.615, 0.959)	**0.0195**
Dominant model				
GA + AA	212 (68%)	237 (75%)	0.717 (0.508, 1.014)	0.0595
GG	101 (32%)	81 (25%)	1	
Recessive model				
AA	57 (18%)	78 (25%)	0.685 (0.467, 1.006)	0.0530
GA + GG	256 (82%)	240 (75%)	1	
Sun et al. (China) (*n* = 1051 + 1169 = 2220)				
GG	356 (34%)	344 (29%)		
GA	517 (49%)	592 (51%)		
AA	178 (17%)	233 (20%)		
G	1229 (58%)	1280 (55%)	1	
A	873 (42%)	1058 (45%)	0.859 (0.763, 0.968)	**0.0125**
Dominant model				
GA + AA	695 (66%)	825 (71%)	0.814 (0.680, 0.974)	**0.0244**
GG	356 (34%)	344 (29%)	1	
Recessive model				
AA	178 (17%)	233 (20%)	0.819 (0.660, 1.016)	0.0696
GA + GG	873 (83%)	936 (80%)	1	

A *p*-value ≤ 0.05 is statistically significant and is shown in Bold.

## Data Availability

The data presented in this study are available in the Appendix A.

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
