# Peer review of "Fibroblast Growth Factor 20 Gene Polymorphism in Parkinson’s Disease in Asian Population: A Meta-Analysis"

_genes, 2021, doi:10.3390/genes12050674_

Round 1

Reviewer 1 Report

The present study shows a PD risk-loci (rs591323) genotyping in a cohort of 586 patients and 586 healthy controls. This genotyping result is used together with two filtered works to examine the association of the SNP with the risk of PD. The genotyped SNP has previously been studied in many populations and cohorts and the present study confirms the protective role of the minor allele (A) in PD.

Minor comments:

Line 39: parenthesis for SNP reference is no needed and therefore, confusing.

Line 58: “primers” need to be correctly spelled

Line 69: “from” need to be correctly spelled

Table1: initials like “HWE” should be specified what they stand for.

Mayor comments:

The genotyping of the samples is done with iPLEX technology. Could the authors explaine why this method was used? I believe it is useful for the screening of many markers, but as the present work is focused in a single SNP, an explanation would be clarifying.

In the introduction, many SNPs are said to be PD risk loci, but the present work is focused in just one locus. It would be of help to explain why and if this is related to the previous comment regarding methodology.

Line 43: It would be helpful for the understanding of the paper to make explicit that the meta-analysis is done with one study of own results (new), and two previous papers.

This should be also explained in the abstract. It is somehow confusing to read that a “total” of 586X2 samples were included, without naming how many Total samples including de 2 papers for the meta-analysis.

Line 52: Controls are said to be matched for gender and ethnicity, but age might also be important as PD is predominantly a late onset disease and therefore, young controls should be avoided, as they could easily convert in following years.

Reviewer 2 Report

Review of a manuscript “Fibroblast growth factor 20 gene polymorphism in Parkinson’s disease in Asian population: a meta-analysis” by Han-Lin Chiang and coauthors submitted to “Gene”.

Parkinson’s disease is a severe neurodegenerative disorder second in prevalence after Alzheimer’s disease. There is no efficient treatment changing the course of the disease and no early biomarkers indicating the beginning of changes on early steps of the illness. Therefore, investigating of genetic components and biochemical pathways is a must to better understand molecular components of this disorder. The authors of the manuscript analyzed the association of rs591323 with the risk for Parkinson’s disease in a Taiwanese population and conduct a meta-analysis to examine the role of this SNP in Taiwanese/Chinese population. This is an important field of study and the results presented in the manuscript will be interesting for “Gene” readers.

The following corrections should be made.

Abstract.

In the Abstract the authors use twice Taiwanese/Chinese populations and one time Taiwanese  populations. It is not clear whether they do it on purpose or by mistake. Also Taiwanese/Chinese population is written in the Abstract one time in singular and the other in plurals. The authors should try to be consistent.

Introduction

Lines 27-28. After “Parkinson’s disease (PD) is a neurodegenerative disease with the pathologic hallmark of dopamine neuron loss in the substantia nigra pars compacta (SNpc) [1]” the authors should add a reference on a more recent review “Emamzadeh  FN et al. Parkinson’s disease: Biomarkers, Treatment, and Risk Factors .  Front. Neurosci., 30 August 2018, 12:612 | https://doi.org/10.3389/fnins.2018.00612

Lines 29-31: After “α-synuclein pathology, including Lewy bodies and Lewy neurites are commonly found in parkinsonian brains, however, the exact role of these abnormal protein aggregations is still elusive and does not correlate with neuronal cell loss [2]” the authors should add a citation on synuclein review: “Synucleins and gene expression: ramblers in a crowd or cops regulating traffic? Frontiers in Molecular Neuroscience, July 2017, 10, 1-7 224 doi: 10.3389/fnmol.2017.00224

Materials and Methods

Line 61: “Taq polymerase, 500 nmol of each PCA primer mix and 2.5 mM of each deoxynucleotide” .The authors should first give the full name of PCA before using it in abbreviated form.  

Line 68: ” A cation exchange resin was used to remove residual salt form the reactions.” The authors should give details of this procedure so it may be reproduced.

Line 82: ”…and that there was sufficient data including genotype frequency for data analysis” should be corrected as follows :”… and if there was sufficient data including genotype frequency for data analysis.”

Results

Line 102:”In dominant model, the AA+GA genotype frequency is significant higher in HC…” This should be rewritten as “In dominant model, the AA+GA genotype frequency is significantly higher in HC…”

Lines 114-115: “A total of 148 publications were included in the initial search. Of these, 62 were duplicates.” It is not clear why mention here the duplicates. If these are just the same publications repeated more than one time, it is better just not to include them and do not mention them at all.

Discussion

Line 137: “The study conducted by Sun and colleagues also showed that both the A allele and the dominant model (AA+AG vs. GG) of rs591323 is protective against PD” This sentence should be corrected as “The study conducted by Sun and colleagues also showed that both the A allele and the dominant model (AA+AG vs. GG) of rs591323 are protective against PD”

Line 141:” In support of the previous studies,” the authors should give citation of these previous studies.

Lines 147-148. “Fibroblast growth factor-20 (FGF20) is one of the neurotrophic factors that was found to selectively expressed in the SNpc [13] and promotes dopamine neuron differentiation and survival[8].” This sentence should be corrected as follows :”FGF20 is one of the neurotrophic factors that was found to be selectively expressed in the SNpc [13] promoting dopamine neuron differentiation and survival[8].”

Round 2

Reviewer 1 Report

In this second revision of the work, minor suggestions have been corrected, but not all of the major recommendations have been answered or not sufficiently explained.

# The authors have explained the iPLEX methodology extensively and have included the following explanation in the methods:

We used iPLEX technology to genotype 20 PD risk foci. ”for genotyping of the 20 SNPs” in the text.” (lane 58)

But nothing is said in the introduction nor results about these 20 analyzed SNPs. Why genotype 20 SNPs if the work is focused in just one?.

Last sentence of the “Genetic analysis” (lane 83) should not be in M&M without an extended analysis and should be argued.

# Regarding control samples age, supplementary data show that age between groups is endeed stadistically (not statically!! As it is written by the author) different, so lane 191 should be removed. But most important is that young controls (more than 100) are no good controls for PD as the authors mentions in lane 192.
